# D-α-Tocopherol-Based Micelles for Successful Encapsulation of Retinoic Acid

**DOI:** 10.3390/ph14030212

**Published:** 2021-03-04

**Authors:** Guendalina Zuccari, Sara Baldassari, Silvana Alfei, Barbara Marengo, Giulia Elda Valenti, Cinzia Domenicotti, Giorgia Ailuno, Carla Villa, Leonardo Marchitto, Gabriele Caviglioli

**Affiliations:** 1Department of Pharmacy (DiFAR), University of Genoa, Viale Cembrano 4, 16148 Genova, Italy; baldassari@difar.unige.it (S.B.); alfei@difar.unige.it (S.A.); ailuno@difar.unige.it (G.A.); villa@difar.unige.it (C.V.); caviglioli@difar.unige.it (G.C.); 2Department of Experimental Medicine—DIMES, University of Genoa, Via Alberti L.B. 2, 16132 Genova, Italy; barbara.marengo@unige.it (B.M.); giuliaelda.valenti@edu.unige.it (G.E.V.); cinzia.domenicotti@unige.it (C.D.); 3Department of Sciences for the Quality of Life, University of Bologna, Corso D’Augusto 237, 47921 Rimini, Italy; leonardo.marchitto@unibo.it

**Keywords:** retinoic acid, micelles, TPGS, nanocarriers, drug delivery systems, topical application, nanocarrier-loaded gels

## Abstract

All-*trans*-retinoic acid (ATRA) represents the first-choice treatment for several skin diseases, including epithelial skin cancer and acne. However, ATRA’s cutaneous side effects, like redness and peeling, and its high instability limit its efficacy. To address these drawbacks and to improve ATRA solubilization, we prepared ATRA-loaded micelles (ATRA-TPGSs), by its encapsulation in D-α-tocopheryl-polyethylene-glycol-succinate (TPGS). First, to explore the feasibility of the project, a solubility study based on the equilibrium method was performed; then, six ATRA-TPGS formulations were prepared by the solvent-casting method using different TPGS amounts. ATRA-TPGSs showed small sizes (11–20 nm), low polydispersity, slightly negative zeta potential, and proved good encapsulation efficiency, confirmed by a chemometric-assisted Fourier transform infrared spectroscopy (FTIR) investigation. ATRA-TPGS stability was also investigated to choose the most stable formulation. Using Carbopol^®^ 980 as gelling agent, ATRA-TPGS-loaded gels were obtained and analyzed for their rheological profiles. Ex vivo release studies from ATRA-TPGSs were performed by Franz cells, demonstrating a permeation after 24 h of 22 ± 4 µ cm^−2^. ATRA-TPGSs showed enhanced cytotoxic effects on melanoma cells, suggesting that these formulations may represent a valid alternative to improve patient compliance and to achieve more efficacious therapeutic outcomes.

## 1. Introduction

Topical all-*trans*-retinoic acid (ATRA) is extensively used as a bioactive molecule in the treatment of inflammatory and proliferative skin diseases such as psoriasis, acne, actinic keratosis, photoaging and epithelial skin cancer. In particular, in cutaneous malignancies, by promoting the differentiation of cancer cells towards their normal phenotype, ATRA represents a valid tool for counteracting cancerous lesions [1]. In addition, ATRA has already marked an important advance in the treatment of acute promyelocytic leukemia, and in vitro studies proved its efficacy in non-small cell lung carcinoma, prostate carcinoma, melanoma, neuroblastoma, etc. [1,2]. Recently it was demonstrated that a topical application of ATRA could inhibit melanoma growth, not only by exerting a differentiating effect, but also by influencing tumor immunity by up-regulation of chemerin, which in turn enhances Chemochine Like Receptor-1 (CMKLR-1)-expressing natural killer cell infiltration, creating an immunosuppressive tumor environment [3,4]. Although melanoma represents only 4% of skin cancers, because of its high aggressive behavior and poor prognosis, it is responsible for 80% of deaths related to dermatological cancers [5]. However, the complex mechanism of action of retinoids is still far from being completely understood [6]. Collectively, the potential clinical applications of ATRA, both as a mono-therapeutic compound or in combination with other drugs, could be extremely wide and not only related to the interaction with the retinoid receptor. Specifically, in acne treatment, ATRA associated with clindamycin or benzoyl peroxide in gel formulations leads to a greater mean total lesion-count reduction than single drugs. However, the adverse effects are still remarkable, including irritation, dryness, exfoliation, erythema, pruritus, burden sensation, and dermatitis. These symptoms, which have been observed in about 85% of ATRA treated subjects, contribute to the poor adherence of the medication [7]. Moreover, ATRA efficacy is strongly limited by several disadvantages, including low water solubility, rapid catabolism by cytochrome P450 enzymes, and high instability in the presence of oxygen, light, and heat [7,8]. Current ATRA formulations, present in the market, mainly consist of aqueous solutions mixed with organic solvents, such as ethanol and propylene glycol (i.e., Effederm^®^), or cream (RetinA^®^). However, traditional dosage forms such as cream, gel, and ointment are not efficient for topical treatment, because of the poor penetration of the drug into the targeted layer of the skin. In an attempt to enhance ATRA efficacy and safety, a microparticulate carrier (Retin-A Micro^®^) reached the market in 1997. Nevertheless, its inferiority versus nanocarriers has been proved by the recent advantages of the nanotechnologies. Indeed, nowadays, the most accredited strategies rely on the employment of nanotechnological systems able to encapsulate drug molecules, thus minimizing the undesirable reactions of the drug in the skin. At the same time, these drug delivery systems are capable of improving drug absorption through the stratum corneum by several mechanisms, including the increment of drug concentration in the formulations, the increase of skin lipids fluidity, and the promotion of drug partitioning into the skin. It was proved that carriers with size >600 nm lay on the skin surface, vesicles of about 300 nm penetrate the stratum corneum, while only >70 nm carriers are able to reach viable epidermal and dermal layers [9]. Until now, several nanocarriers have been investigated for ATRA topical application, including liposomes, niosomes, solid lipid nanoparticles (SLNs), nanostructured lipid carriers (NLCs), cyclodextrins, and polymeric micelles [10,11,12,13,14,15]. In this regard, liposomes represent one of the most studied systems [10]. The commonly used technique to prepare liposomes for skin delivery is thin-film hydration, which often starts from the solubilization of lipids in hazardous organic halogenated solvents. In addition, in this case, the lipophilic drug accumulated in the phospholipid bilayer instead inside the aqueous core, affecting system stability over time. Nevertheless, encapsulation efficiency, large vesicle size, and scale-up issues represent challenges yet to be overcome [9]. Niosomes show an improved resistance to oxidation; however, current research on these vesicles is rather limited [11]. Compared to lipid-based vesicular carriers, ATRA-loaded SLNs are currently preferred for their scalable manufacturing process, smaller particle size, major protection of the drug, and good drug-loading capacity. Despite these improvements, SLNs can undergo drug expulsion during storage, and no data confirm NCLs long-term physical stability [12,13]. Due to the presence of a rigid cyclic structure, with hydroxyl groups oriented on the external surface, cyclodextrin is able to permeate lipophilic biological barriers, such as the skin [14]. Recently, ATRA micelles composed of amphiphilic di-block methoxy-poly(ethylene glycol) di-(hexyl-substituted lactic acid) (MPEG-di-hex-PLA), expressed preferential deposition into the pilosebaceous unit (PSU) compared to the free ATRA solution, confirming that the PSU represents the main penetration route into the skin, only for nanoparticulate vehicles [15]. Even if only a few studies are present in the literature, it seems that polymeric micelles localize in different layers of the epidermis, providing a slow and controlled drug release from intact micelles [16].

Here, for its synergistic effect to act as nanocarrier and as permeation enhancer, we chose D-α-tocopheryl polyethylene glycol succinate (TPGS) as micelle-forming material for ATRA encapsulation [17]. TPGS is a water-soluble synthetic derivative of natural vitamin E, containing 260 mg/g of the vitamin. TPGS is obtained by esterifying vitamin E succinate with polyethylene glycol (PEG) 1000, and, in recent years it has been successfully used to develop efficacious drug delivery systems [18]. Since TPGS is a waxy solid with a melting point of 37–41 °C and remains stable up to 199 °C, it has particularly attracted the interest of the pharmaceutical industry for this resistance to high temperatures without deterioration. In addition, due to its amphiphilic nature, TPGS is capable of self-assembling in water above its critical micellar concentration (CMC), equal to 0.02 wt%, in nanosized aggregates of about 13 nm. In this context, TPGS has been widely studied as an absorption and permeation enhancer, emulsifier, and solubilizing agent [17,18]. Moreover, TPGS may exert intrinsic anticancer effects, without inducing toxicity to normal cells, and its capability to trigger apoptosis in prostate, lung, and breast carcinomas has already been demonstrated [19]. Although these activities have been extensively studied, the intracellular signaling pathways involved in the induction of its pro-apoptotic action are not completely understood [20]. Indeed, like other nonionic surfactants, TPGS, administered at a concentration of 0.1 mg/mL (below its CMC), can reduce the activity of the P-glycoprotein (P-gp), a member of the ATP-binding cassette (ABC) transporter family involved in the acquisition of multidrug resistance (MDR) in cancer cells [21,22]. Briefly, by binding the mitochondrial respiratory complex II, TPGS leads to a reduction in intracellular ATP levels, and consequently, to the inhibition of the ATPase-dependent activity of P-gp, favoring the intracellular accumulation of chemotherapeutics [22]. Moreover, recently, it was reported that TPGS could also directly inhibit P-gp by preventing ATP hydrolysis [23]. Therefore, in anticancer drug formulations, TPGS may exert a dual function as a solubilizing delivery system for poorly soluble anticancer drugs, as well as a direct cytotoxic agent. As a solubilizing agent, TPGS may also improve the stability of both drugs and lipid-based nanoparticle carriers, which are particularly susceptible to oxidation [24].

In this work, we approached the problem of the main drawbacks associated with the topical administration of free ATRA by designing and optimizing the encapsulation of ATRA in different TPGS amounts. Until now, TPGS in topical formulations has been used for minoxidil [25], estradiol [26], and griseofulvin [27] as a penetration enhancer and mixed with a cosolvent, such as ethanol and PEG 400. However, in these studies, the micelles were not fully developed and characterized, and TPGS was mainly employed to increase drug permeability, in a context where drug solubility was improved by the presence of the cosolvent. Here, micelles loaded with ATRA (ATRA-TPGSs) were obtained and characterized by Fourier transform infrared spectroscopy (FTIR), thermal analysis, photon correlation spectroscopy (PCS) for particle size, and polydispersity index (PDI) and Zeta (ζ) potential determination. A selected ATRA-TPGS colloidal suspension was further loaded in a hydrogel matrix, which represents the ideal vehicle for topical drug delivery, both for its feasibility and patient compliance. Indeed, nanocarrier-loaded hydrogels, made of Carbopol^®^ 980 (carbomer) as the gelling agent, combine the advantages of a standard technology with the improvements offered by nanotechnology [28]. A comprehensive sketch of the chemical structures and ATRA-TPGS dispersion into the gel polymer network is reported in Figure 1.

Carbopol^®^ 980 is frequently selected for the preparation of hydrogels because of its ability to provide high viscosity at low concentration and its compatibility with many active ingredients [29]. A rheological characterization was performed on ATRA-TPGS hydrogels to assess their flow behavior, and an ex vivo permeation study was carried out on ATRA hydro-alcoholic solution, micellar dispersion, and gel to determine ATRA transdermal permeation. Finally, the cytotoxic effects of an ATRA-TPGS freeze-dried suspension were evaluated on a melanoma cell line. In our goals, ATRA encapsulation into a vitamin E micellar core would have allowed it to increase its water solubility, stability, and thus efficacy, providing at the same time a safer administration by avoiding the use of toxic dimethyl sulfoxide (DMSO) or of other cosolvents, such as ethanol and propylene glycol.

## 2. Results

### 2.1. Effect of TPGS on ATRA Solubility

ATRA is a weak acid and its solubility in water is 0.19 μg/mL (6.3 × 10^−4^ mM) [30]. Before carrying out the preparation of ATRA-TPGSs, studies concerning the suitability of TPGS as an encapsulating agent were performed. For this purpose, the micelles were firstly prepared by the equilibrium method. These studies were carried out by adding increasing amounts of TPGS to aqueous saturated ATRA solutions. The mixtures were kept at a fixed temperature under stirring for the time necessary for equilibrium to be reached during the process of complex formation. The concentration of ATRA at equilibrium was determined by UV-Vis spectrophotometric analysis using a constructed linear calibration curve Equation (1), Appendix A. As reported at https://www.statology.org/good-r-squared-value/ (Accessed on 3 March 2021), in scientific studies, the *R*-squared may need to be above 0.95 for a regression model to be considered reliable, so the high value of the correlation coefficient (*R*^2^ = 0.9904) related to the curve assured linearity within an ATRA concentration range from 5.33 × 10^−3^ to −2.13 × 10^−2^ µM.
(1)y=143.21x−0.02

The obtained results showed the considerable effect of TPGS on ATRA solubility. At the maximum TPGS concentration tested, the total ATRA solubility reached the value of 90.76 ± 3.24 µg/mL (about 0.03 mM) (Figure 2a), establishing an improvement by 478-fold. This important enhancement reflects the ability of TPGS to self-assemble in water, leading to supramolecular structures able to host the poorly soluble drug molecules. The concentration of 0.13 mM, found in this phase-solubility diagram, may rationally correspond to the CMC of ATRA-TPGS micelles, and its value is very similar to that reported in literature for empty TPGS micelles (0.2 wt%, 0.132 mM) [31]. Indeed, TPGS does not exert any influence at the lowest concentrations (<0.13 mM), while above its CMC, the solubility of ATRA grows linearly with TPGS concentration (Figure 2b).

Undoubtedly, at concentrations of TPGS greater than its CMC, the total (namely, the free plus encapsulated ATRA) drug concentration in the solution is directly proportional to the concentration of the added complexing agent and reaches the maximum value of 0.301 mM. CMC mainly depends on the composition of the hydrophobic segments forming the micelle core, and it is related to the stability of the supramolecular aggregates. Low CMC values indicate a lower tendency of micelles to dissociate when diluted in the bloodstream. Since TPGS CMC is relatively high (10^−4^ M), when parenteral administration is needed, TPGS should be associated to another surfactant to reach CMC values close to 10^−5^ or 10^−6^ M [32,33]. Only one study reported the superior advantages of TPGS micelles by their intramuscular administration to mice [34]. Nowadays, TPGS has been widely recognized as a valid tool to improve solubility and permeation in oral, transdermal, and topic drug delivery systems [35].

### 2.2. Preparation and Characterization of ATRA-Loaded TPGS Micelles (ATRA-TPGSs)

For ATRA incorporation into TPGS micelles the solvent-casting method was preferred. The two preparative methods employed, shake flask and solvent casting, are indicative of two different routes of drug encapsulation. In the first one, the drug and the micelles are added simultaneously in a water solution, while in the second one, the drug is introduced into the system before micellization. Due to ATRA’s poor aqueous solubility, the solvent-casting method seemed the be more suitable to reach a good drug encapsulation efficiency with a faster kinetic. After the evaporation of the ethanol, an opalescent yellow wax film formed, in which drug molecules were tightly entangled with TPGS. The addition of a hydration solution induced the PEG chain to associate with water molecules through hydrogen bonding, forming the shell, while the vitamin E moiety attracted ATRA molecules inside the micellar core. Moreover, by varying the weight ratio of the drug (ATRA) to the polymer (TPGS) in the preparative mixture, we examined the best encapsulation conditions by determining the changes in the total ATRA solubility [36]. In particular, six formulations were prepared with a constant ATRA concentration of 1 mg/mL, but with different increasing TPGS amounts, ranging from 20 to 70 mg/mL (Figure 3).

After hydration of the film with water, all the formulations from the ratio 1:50 to 1:70 appeared quite limpid and very yellowish, while at lower ratios the colloidal suspensions appeared very cloudy and needed to be filtered prior to further characterization (Appendix A). The measurements of total solubility showed that the amount of ATRA in the solution increased according to the TPGS addition from 0.66 ± 0.08 mM to 1.7 ± 0.2 mM. This was possible because, by increasing the concentration of TPGS, a greater number of micelles can establish; thus, more ATRA molecules can pass from the solid state to the encapsulated form. The formulations from 1:50 to 1:70 were able to incorporate enough ATRA to reach drug contents of about 0.43 mg/mL and 0.53 mg/mL, respectively, the latter corresponding to a 2789-fold increase in ATRA aqueous solubility.

The ATRA encapsulation efficiency (EE%) and the drug loading capacity (DL%) associated to each formulation are reported in Table 1. From the experimental data, obtained by spectrophotometrically assessing the concentrations of ATRA in the micellar dispersions, it was observable that the EE% enlarged from 35 ± 2% to 79 ± 8% as the ATRA:TPGS ratio increased, confirming the suitability of the method employed. The EE % is supposed to increase with increasing TPGS volume fractions and TPGS hydrophobic head interactions with ATRA. However, between the 1:50 and 1:60 ratios there was only a small improvement in the corresponding EE, and consequently, in the ATRA solubility increase. Drug encapsulation depends on the partitioning of drug into the micelle core, which is correlated to TPGS concentration; however, a proportional progression is not always present overall at higher EE values which may raise different drug–polymer and polymer–polymer interactions [37]. Concerning the DL%, it was observable that it ranged from 0.8% to 0.9%. Drug loadings were relatively low, but they were similar to data previously reported in other works [15]. The explanation relies in the interactions that may take place between the drug and the vehicle. The major driving forces that may lead to a strong association are as follows, in decreasing order: number of H-donors, intermediate log *p*, number of H-acceptors, and aqueous solubility [38]. ATRA may display few H-donors and H-receptors, a log *p* of 6.2, and an overall ionizable site, and all these features negatively impact its incorporation into micelles.

Regarding the particle size, the analysis by PCS was performed both on the freshly prepared micellar dispersions and on the lyophilized samples upon reconstitution (Table 2, Appendix A). All the freshly prepared micellar formulations showed homogenous nanometric dimensions, with hydrodynamic diameters ranging from 11.4 ± 0.1 nm to 17 ± 5 nm (Table 2, Appendix A). As expected and reported, ATRA-TPGSs deriving from lower drug-to-polymer ratios showed a slight reduction of the mean diameters, suggesting a better drug-vitamin E packing into the micelle core and a slighter dehydration [39,40]. In this regard, although studies reported that values of particle size were not correlated to the TPGS concentration [41], a generally inverse correlation can be observed between the amount of TPGS used and particle size. Concerning our results, they demonstrated that such a correlation exists, except for the data obtained from the micellar formulation with an ATRA:TPGS ratio of 1:30, which, on the basisof its high standard deviation, could be considered unreliable. Moreover, all these formulations showed a unimodal distribution. The PDI value in the analyzed samples fluctuated between 0.1 and 0.3, indicating the presence of a monodisperse distribution of the particle population, which is required for more reproducible in vitro and in vivo results. As expected, the micelles subjected to the freeze-drying process showed slightly greater dimensions and PDI values than the fresh ones (Table 2 and Appendix A) because of their aggregation tendency during the removal of the solvent [42]. Nevertheless, no overall significant variations between the freeze-dried and fresh formulations were evidenced, except for at the ATRA:TPGS ratio of 1:60. The ƺ potential is a useful parameter for predicting the stability of nanoparticulate dispersions and their tendency to aggregate. Here, we investigated if a net electrical charge was present on the micelles surface in two different conditions, i.e., with the micelles dispersed in milli-Q water or in a 5 mM 4-(2-hydroxyethyl)-1-piperazineethanesulfonic acid (HEPES) buffer (pH 7.4) (Table 2, Appendix A). Although milli-Q water was later chosen as the preferred medium for ATRA-TPGS preparations, it was considered mandatory to establish what could happen to the micelles in a physiological medium. Consequently, the size and the ƺ potential were also detected in HEPES buffer. The mean diameter was not affected by the change of the pH, while the ƺ potential was surprisingly slightly negative in milli-Q water, and almost neutral in the saline solution where the neutralization of the charges may occur. Since TPGS does not possess ionizable groups, its surface charge is almost neutral (data not shown), so the negative ƺ potential could be addressed to ATRA-dissociated carboxyl groups, which probably cannot be internalized into the micelles core, remaining exposed on the surface, and thus inducing ion rearrangement around micelle micro-environment [43]. Similar to what was reported in a recent study, the samples measured in water had an average ƺ potential value of −10.5 mV [43]. Although the ƺ potential values were weakly negative, the micelles stabilization occurred mainly by steric hindrance, and not electrostatic repulsion, due to the presence of the PEG chains, which determined the formation of a solvation shell around the micelles, preventing them from aggregating. As expected, the mean value detected in the samples analyzed in the 5 mM HEPES buffer was 2.9 mV (Table 2), because in the saline solution the charge of the particles was neutralized. Additionally, in this case, for the stabilization of the micelles and to avoid their aggregation, the presence of the PEG backbone was crucial.

### 2.3. FTIR Analysis Assisted by PCA

FTIR allows for rapid analytic data that could provide chemical information about manufacturing processes, sample quality, and differences or similarities in chemical composition [44]. In this work, the FTIR analyses were carried on individual components (ATRA and TPGS), and on lyophilized ATRA-TPGSs with a different ATRA:TPGS (*w*/*w*) ratio, to qualitatively evaluate the success of the encapsulation reaction.

#### 2.3.1. FTIR Spectra

Figure 4 shows the FTIR spectra of ATRA, TPGS, and three ATRA-TPGS samples containing ATRA and TPGS in 1:1, 1:5, and 1:10 (*w*/*w*) ratios, purposely prepared to assess if in the formulations with low TPGS content, the typical signals of ATRA were detectable. Another image, showing the FTIR spectra of the six samples containing ATRA and TPGS in 1:20, 1:30, 1:40, 1:50, 1:60, and 1:70 ratios, is available in Appendix A.

Concerning ATRA (light green spectrum), the principal peaks were observed at 1689 cm^−1^ (C=O acid), in the range 3068−3049 cm^−1^ (C=C–H stretching) and in the range 2933−2820 cm^−1^ (C-H stretching of CH_3_ groups). Bands of C=C and C–O stretching were observable at 1603 and 1572 cm^−1^ and at 1253 and 1185 cm^−1^, respectively. O-H stretching was not clearly detectable. As for TPGS (orange spectrum), the main peaks were observed at 1739 cm^−1^ (C=O esters) in the range 3600−3200 cm^−1^ (O–H stretching) and in the range 2923−2870 cm^−1^ (C–H stretching of CH_3_ and CH_2_ groups). Aromatic bands were not clearly detectable. By a simple observation of Appendix A, all the ATRA:TPGS spectra were very similar to one another and to that of TPGS, and very different from that of ATRA. Sometimes, it is possible to compare the spectral data and detect the physicochemical differences among the samples under study, simply by observing the spectra recorded. However, this is a case in which it was impossible to state if the samples contained ATRA by simple observation of the FTIR spectra. In such situations, an analytical toolthat works on the complex matrix of spectral data obtained by all samples is necessary. The analytic method allows for identifying, among the sample population, clusters indicating unequivocally which compound (ATRA or TPGS) the samples under study (ATRA-TPGSs) are more similar to in their composition on the basis of their scores plot.

Multivariate analysis (MVA) is a rapid and efficient chemometric tool suitable to study complex instrumental data sets and able to unveil the hidden information in them [45]. There are more than 20 different ways to perform MVA depending on the type of data and the objectives to be achieved, including principal component analysis (PCA). PCA is the most widely used chemometric technique for handling FTIR spectral data and obtaining the desired information.

#### 2.3.2. Principal Component Analysis (PCA)

PCA, a method widely used in MVA, particularly with spectral data consisting of thousands of variables that necessitate data reduction, was used in the present study for our exploratory purposes concerning the presence of ATRA in the micelles.

In PCA, multidimensional data are reduced to a small number of new variables, called principal components (PCs), which are orthogonal linear combinations of the original ones that efficiently represent data variability in low dimensions. The information carried out by PCs is expressed as a percentage of the explained variance. By definition, PC1 has the largest % explained variance, followed by PC2, PC3, and so on [46,47]. PCA may identify important spectral regions that differentiate the samples belonging to a test population and is capable of creating clustering patterns on the basis of the spectral data that depend on the physicochemical characteristics of the samples. PCA provides score plots, where one component (e.g., PC1) is displayed versus another (e.g., PC2), and where the samples under study assume specific positions (scores), forming groups of similar compounds. The position taken by each sample on the selected component provides reliable predictive information on its physicochemical characteristics.

In addition, PCA results can also be represented as loading plots. The loading plot graphs the coefficients (loadings) of each variable considered in the PCA (in our case the variables were the wavenumbers in the range 4000−1000 cm^−1^) for the first component versus the coefficients for the second component. The loading plots allow us to identify which variables, i.e., which wavenumbers, have the largest effect on each component. Loadings can range from −1 to 1. Loadings close to −1 or 1 indicate that the variable strongly influences the component. Loadings close to 0 indicate that the variable has a weak influence on the component. Evaluating the loadings can also help characterize each component in terms of the variables. The results of PCA allowed us to cluster the samples with a higher content of ATRA separately from those containing less amounts of ATRA, and to visualize their positions with respect to those of pure ATRA and TPGS. Figure 5 shows the PCA results represented both as a score plot (Figure 5a) and a loading plot (Figure 5b).

Accordingly, on the principal component 2 (PC2), all ATRA-TPGSs were found to be located in the part of the score plot with negative scores including the ATRA scores (Figure 5a), and well separated from the TPGS scores, which were located at positive scores in both PC2 and PC1. These results asserted the existence of chemical similarities between ATRA-TPGSs and ATRA, thus confirming the presence of ATRA in the prepared micelles. In addition, the population of ATRA-TPGSs resulted in further separation into two subgroups on the PC1. The group of samples that was appositively prepared for the FTIR analysis and had a higher content of ATRA was located very close to ATRA, while the group including the samples mentioned in Section 2.2 and named 01–06, containing lower ATRA amounts, formed a cluster including TPGS that was observable at positive scores in PC1.

Interestingly, the loading profiles on PC1 and PC2 (blue and red lines, respectively, in Figure 5b) consistently showed that the FTIR bands strongly involved in explaining the sample separation included the bands in the ranges 1800−1600 cm^−1^, 3750−3250 cm^−1^, and 3000−2800 cm^−1^, which are significant areas of the FTIR spectra of both ATRA and TPGS.

### 2.4. Thermal Analysis by Differential Scanning Calorimetry (DSC)

The study of thermal properties of drug and excipient mixtures is crucial in pharmaceutical technology, as it allows us to obtain information about phenomena of melting, recrystallization, or decomposition, which in turn can help to assess the status of the entrapped drug (molecularly dispersed or crystallized) and the interactions among the components that have arisen during the production process [48].

In this work, we studied the thermal behavior of lyophilized drug-loaded formulations of ATRA:TPGS at 1:50 and 1:20 (*w*/*w*) ratios as representative micellar compounds of all those prepared, in comparison to the corresponding physical mixtures, as well as to pure ATRA and TPGS. As depicted in the thermograms (Figure 6), ATRA showed a slight endothermic peak at about 146 °C and a relevant peak associated to melting at about 183 °C [49]. TPGS alone gave an endothermic sharp peak of melting at around 43 °C and a broad peak at 32 °C, due to the minor presence of an amorphous solid [50]. The DSC thermograms of ATRA-loaded micelles showed only the melting peak of TPGS. Interestingly, although not containing the encapsulated ATRA, the physical mixture profiles also evidenced the disappearance of the ATRA melting peak, probably due to ATRA *in situ* solubilization into melted TPGS within the polymer melt during the DSC measurement. Nevertheless, the mixture profiles appeared slightly different, since in the physical mixture, the endothermic peak was more similar to that of TPGS raw material, while in the lyophilized mixture the endothermic peak was less evident.

### 2.5. Stability of ATRA-TPGSs

In order to better identify which formulation among the six was the most valuable for further investigations, stability studies were performed by keeping the different ATRA-TPGS aqueous dispersions in an incubator at 25 °C. In this regard, the detection of total ATRA in the solution was measured after 24, 48, and 72 h. Note that, as long as ATRA remained in the micellar core, the colloidal dispersion denoted stability and appeared clear; on the contrary, when instability and ATRA leakage occurred, turbidity and precipitation were observed. As shown in Figure 7, the collective samples with low ATRA:TPGS ratios (1:20, 1:30, and 1:40) were those that showed a decrease in the solubility of ATRA over time, proving to be more unstable. These visually inspected samples showed yellow ATRA particles at the bottom of the tube. On the contrary, the samples in which the percentage of TPGS was higher (1:50 and 1:60) were characterized by a more constant maintenance of the concentration of solubilized ATRA, proving to remain more stable over time; however, the 1:60 ratio seemed to be less resistant to lyophilization. The ratio of 1:70, on the other hand, was characterized by high fluctuations, due to the high concentration of TPGS, which could determine the ATRA supersaturation of the solution at the preparation moment, consequently undergoing drug precipitation over time. Taking into consideration these results, the ATRA-TPGS formulation, prepared from ATRA:TPGS at a 1:50 ratio (*w*/*w*) in the starting mixture, was selected for use in the subsequent experiments as it was the most stable one.

### 2.6. Nanocarrier-Loaded Hydrogel Preparation and Characterization

The patient adherence to topical treatments for a variety of chronic skin conditions, such as psoriasis, acne, and atopic dermatitis, is known to be very poor. Semisolid dosage forms are the most ancient and popularly known pharmaceutical preparations and their continuous use over time relies on their practical handling and good acceptability by the patients, which are important to assure and increase adherence to the therapy. Therefore, the incorporation of drug-loaded nanocarriers in a hydrocolloid to obtain topical formulations represents one of the most applied strategies, as gels are relatively simple to prepare in comparison with creams, easier to apply, and more adhesive when compared to lotions [51]. To make the ATRA-loaded micellar nanocarriers spreadable for topical administration, starting from ATRA:TPGS at a 1:50 ratio, three gel formulations were prepared with Carbopol ^®^ 980 concentrations of 0.5%, 1.0%, and 1.5%. The gels were obtained by directly adding the gelling agent to the micellar dispersion; indeed, when the gel was prepared separately and then joined to the micellar dispersion, a remarkable decrease of viscosity was observed. Additionally, it was possible to add the lyophilized powder to the gel base and stir until a homogenous dispersion was obtained (25 °C, 30 min, 100 rpm). All the gels were light yellow, translucent in appearance, and free of lumps (Figure 8). The pH was measured after neutralization and ranged from 5.0 to 5.8; therefore, it was biocompatible with the skin and at the same time not too high to enhance ATRA carboxyl group dissociation, which could hamper drug permeation through the skin. When observed under optical microscopy, no precipitates were detected inside any of the gel formulations Moreover, the gels were diluted at 1:10 *w*/*w* with water and analyzed by PCS to detect eventual changes in micelles size, but no significant variations were observed.

#### Rheological Studies

Rheology is a fundamental parameter to be considered when a formulation for skin application is prepared. From the data collected performing rheological studies on the prepared nanocarrier-loaded gels, two types of graphs were constructed: one reporting the shear stress (τ [Pa]) as a function of the shear rate ($\gamma^{.}$ [s^−1^]) (Figure 9), and another reporting the viscosity (η [Pa*s]) as a function of $\gamma^{.}$ (Appendix A). In Figure 9, it was observed that τ was not directly proportional to $\gamma^{.}$ for all samples, thus concluding that η was not constant and that the prepared hydrogels have a non-Newtonian behavior. For definition in non-Newtonian models, there is usually a region at both the low and high values of $\gamma^{.}$, where the viscosity is independent or nearly independent of $\gamma^{.}$, and a section in between that exhibits strong values of $\gamma^{.}$ dependence [52]. Accordingly, in Appendix A it can be observed that η was not constant and decreased rapidly up to a value of $\gamma^{.}$ of 50 s^−1^, and less rapidly between 50 and 250 s^−1^. For $\gamma^{.}$ > 250 s^−1^, η no longer changed, becoming independent from $\gamma^{.}$, thus confirming a non-Newtonian behavior for all the hydrogels prepared. Since η decreased as $\gamma^{.}$ increased, our samples showed a shear-thinning behavior. Finally, it was also evident that the η was higher in the gels containing a higher quantity of the gelling agent.

For a semisolid formulation, the rapid decrease in η is a positive finding, because it allows us to apply the gel easily, as it will be increasingly fluid during its spreading on the skin.

To characterize the flow properties and to better describe the τ/$\gamma^{.}$ relationship by determining the index flow (*n*) [52], mathematical models were applied to the data from the rheograms reported in Figure 8. For non-Newtonian fluids, as was our case, the most common mathemathical models include the Power Law (or Ostwald), the Herschel–Bulkley, the Casson, and the Bingham Plastic rheological models, which are expressed by Equations (2)–(5), respectively.
τ = κγ*^n^*(2)
τ = τ_oH_ + κ_H_γ*^n^*^H^(3)
τ^0.5^ = κ_oc_^0.5^ + κ_c_^0.5^γ^0.5^(4)
τ = τ_o_ + η_p_γ(5)
where *n* and *n*H are the fluid flow behavior indexes, which indicate the tendency of a fluid to shear thin or thick, and k and kH are the consistency coefficients, which serve as the viscosity indexes of the systems.

In particular, when *n* < 1, the fluid is shear thinning (pseudo-plastic fluid), when *n* = 1, the fluid is Newtonian, and when *n* > 1, the fluid is shear thickening (dilating hydrogels). τ_oH_ is the Herschel–Bulkley yield stress point, k_oc_ is the Casson yield stress point (τ_oc_), and k_c_ is the Casson plastic viscosity (η_cp_). In the Bingham Plastic model, τ_o_ is the yield stress point and η_p_ is the plastic viscosity. Note that the Hershel–Bulkley model is an extension of the Bingham Plastic model to include shear rate dependence.

According to the literature [52], the Power law and Hershel–Buckley rheograms were obtained by reporting in graph Log τ versus Log γ (Figure 9) and Log (τ-τ_o_) versus Log (γ) (Appendix A), respectively. In both plots, Log k and *n* were obtainable from the relative linear regression equations, where Log k is the intercept and *n* is the slope. The k values were calculated accordingly. The values of τ_oH_ were obtained as reported in the literature [52]. The Casson rheograms were plotted by reporting the square roots of τ versus those of γ in the graph (Appendix A). In the plot, the Casson parameters (τ_oc_ and η_cp_) were obtained from the relative linear regression equations, where k_c_^0.5^ is the slope and k_oc_^0.5^ is the intercept: the Casson yield stress was calculated as the square of the intercept, [τ_oc_ = (k_oc_^0.5^)^2^], and the Casson plastic viscosity was the square of the slope [η_cp_ = (k_c_^0.5^)^2^]. Finally, the Bingham Plastic model describes the behavior of visco-plastic materials, which behave as a rigid body at low stresses, but flow as a viscous fluid at high values of stress. The value of τ, above which they flow as viscous fluids, represents the yield stress point τ_o_. The linear regression equations, in which the yield stress point (τ_o_) is the intercept and plastic viscosity (η_p_) is the slope, were obtained by the rheograms plots reporting τ versus γ (Appendix A). The equations of all the linear regressions obtained by the mathematical models, the correlation coefficients (*R*^2^), and the values of slope and intercept are reported in Table 3. Moreover, in Appendix A a table reporting the model parameters, i.e., *n* and *n*H, k and kH, τ_oc_, η_cp_, τ_o_, and η_p_, is available (Appendix A).

The maximum correlation coefficient has been considered as the statistical parameter to designate the model with the best fit to the data. Accordingly, the mathematical model that best described the rheological behavior of the studied hydrogels was the Power Law (or Ostwald–de Waele) model, which is an easy-to-use model, ideal for describing the behavior of shear-thinning, relatively mobile fluids, such as weak gels and low-viscosity dispersions (Figure 10).

Moreover, since the *n* values were remarkably <1 for all the samples, it can be asserted that 0.5%, 1.0%, and 1.5% hydrogels possess a pseudo-plastic behavior, and that among them, the sample containing Carbopol 1.0% can be considered the best one in terms of consistency, allowing for easy spreading on the skin by applying a light massage.

### 2.7. Ex Vivo Permeation Study

In order to evaluate the ATRA diffusion through the skin, we carried out ex vivo permeation studies, using pig skin and vertical Franz diffusion cells in non-occluded circumstances, as it happens when the patient applies a given formulation onto his own skin, for a long period of 24 h, in order to allow the spectrophotometric determination of the total amount of ATRA permeated. The ATRA-TPGS formulation from the 1:50 ratio contained drug concentrations of 0.36 ± 0.01 mg/mL, suitable for topical use, as most European marketed formulations authorized by the European Medicines Agency (EMA) contain ATRA at a concentration ranging from 0.025% to 0.050% *w*/*w*. The data of ATRA skin permeation are reported in Table 4. No statistical difference was detected between the obtained values, and therefore, between the three different formulations.

An exactly weighed amount of the nanocarrier-loaded gel, 1000 µL of the ATRA-TPG micellar dispersion, and the ATRA hydroalcoholic solution were applied on the stratum corneum, so as to have the same drug concentration.

Previous studies of the topical application of TPGS alone were performed at TPGS concentrations ranging from 0.05% to2%, 1% to 5%, or up to 20% *w*/*w*% for minoxidil, griseofulvin, and estradiol, respectively [25,26,27]. However, these formulations included the addition of ethanol or other organic solvents in the starting mixture or in the final preparation. The presence of the solvent increased TPGS CMC, suggesting that the increase in drug penetration was driven primarly by the presence of ethanol, with TPGS essentially acting as surfactant, instead as a micelle-forming material. Here, we prepared ATRA-TPGS micelles at TPGS concentrations ranging from 0.6 to 4.2 *w*/*w* %. The ATRA-TPGS formulation obtained from the 1:50 ratio, and selected for its higher stability, contained TPGS at 3%. This formulation was applied as such, without the adjunct of ethanol, and provided a cumulative amount of permeated ATRA of near 27 µg/cm^2^ after 24 h. This value was higher than that reported for ATRA delivered from niosomes [11] and liposomes [53], probably because the lipophilic materials only reach the upper layer of the skin, where they undergo fusion with the lipid constituents of the skin surface.

Concerning micelles, recent studies have focused on micelle mechanisms of interaction with the skin, but the fate of the micelles upon contact with the stratum corneum is still not completely elucidated [54]. Indeed, it is not clear whether the micelles with their hydrophilic shell go towards disassembled structure or remain intact. Surely, TPGS enhances drug flux by decreasing the interfacial tension and acting as a surfactant. But in addition, it could also act as a drug reservoir, thus exerting a slow drug release from the unbroken micelles, as was proposed for polymeric micelles. Finally, it could also increase drug penetration up to deeper layers of the skin because of its very small size [16].

As already reported, ATRA tends to accumulate in the stratum corneum and only small quantities manages to permeate the receptor compartment. As shown in Table 4, the drug incorporation in the gel is known to result in a slower skin permeation rate than in fluid state vehicles. Besides, the alcoholic solution containing free soluble ATRA molecules provided the greatest permeated amount; however, the presence of the organic solvent represents an insurmountable drawback to a safer drug formulation.

### 2.8. Cytotoxicity Studies Performed on Melanoma Cell Lines

Before studying the biological activity of ATRA-TPGSs on melanoma cells, a preliminary analysis was performed to evaluate the dose-dependent activity of free ATRA. Melanoma cells were treated for 72 h with increasing concentrations of ATRA, ranging from 0.1 to 20 µM. Figure 11 reports the dose-dependent cytotoxic effect of ATRA on melanoma cells.

As reported in Figure 11, cell viability was significantly reduced starting from 5 µM of ATRA, which determined a decrease of 10%. Higher doses further reduced the viability of cells treated with 7.5 and 20 µM ATRA by 20% and 60%, respectively. Since the 20 µM dose was extremely toxic for melanoma cells, inducing necrosis, and given that ATRA, at the dose of 13.2 µM, is cytotoxic for normal skin fibroblast-like cells [55], subsequent studies were performed using ATRA at 5 µM and 7.5 µM. The cytotoxic activity of ATRA-TPGSs was assessed using the lyophilized ATRA-TPGS formulation obtained using the ATRA:TPGS ratio of 1:50 (*w*/*w*), which was also used to prepare ATRA-TPGS hydrogels, as explained above. Melanoma cells were exposed for 72 h to 5 and 7.5 µM free ATRA, to ATRA-TPGSs at a dose that provided ATRA 5 and 7.5 µM, and to TPGS at the same concentrations of those present in ATRA-TPGS (Figure 12). ATRA-TPGSs and TPGS powders were solubilized in the culture medium, while free ATRA was solubilized in DMSO prior the dilution in the medium. Cells with the same volume of DMSO were used as a control.

As shown in Figure 12, after 72 h of treatment, ATRA-TPGS micellar formulation significantly improved ATRA activity at both ATRA-tested concentrations of 5 µM and 7.5 µM. In particular, the encapsulated ATRA at the highest concentration induced a significant cytotoxic effect (*p* < 0.01), both versus the control cells and versus pure ATRA (*p* < 0.05). Additionally, TPGS alone had a partial effect on the viability of melanoma cells, suggesting an additive effect of the two substances, and thus determining a cytotoxicity of ATRA-TPGS significantly higher than that of either ATRA or TPGS alone. Indeed, it was shown that TPGS is able to harden cell membranes and to inhibit the P-gp ATPase, the P-gp energy source of active transport, by acting on the cell surface through its allosteric modulation. Besides, it has been reported that TPGS has an intrinsic anticancer activity capable of determining cancer cell apoptosis, ROS generation, and growth inhibition, whose mechanisms have yet to be investigated [56]. Moreover, it can also be speculated that by inhibiting the overexpression of proteins of the ATP-binding cassette transporter family in tumor cells, which mediate the drug efflux, the presence of TPGS could limit the extrusion of ATRA out of the cell and extend its residence time, thus improving ATRA cytotoxic effects [57]. Furthermore, as in the case of other hydrophobic drugs, the remarkable improvement of ATRA solubility due to ATRA encapsulation in TPGS surely contributed to enhancing its cytotoxic activity, as already observed in our previous study [36]. These findings are particularly interesting since ATRA, in the encapsulated form, is extremely toxic for melanoma cells at concentrations far lower than 13.2 µM, opening new opportunities to treat melanoma cells without affecting normal skin fibroblasts.

## 3. Materials and Methods

### 3.1. Materials

The TPGS was a gift from PMC Isochem (Vert Le Petit, France); the Carbopol^®^ 980 NF was purchased from Lubrizol (Brussels, Belgium); and the ATRA and all reagents were purchased from Sigma Aldrich (Milano, Italia).

### 3.2. Solubility Studies

The equilibrium solubility of ATRA was determined by the shake-flask method. An excess amount of the drug was added in the TPGS water solution and maintained in a water bath under vigorous shaking at 37 ± 0.2 °C by RCS 6 Lauda Thermostat. The TPGS concentrations ranged from 0.05 to 3.30 mM. The samples were kept in the dark and allowed to equilibrate until the ATRA concentration in solution remained constant. After 48 h under stirring, the samples were filtered using a 0.22 µm filter (Minisart RC Sartorius, Göttingen, Germany. Aliquots of each filtrate were diluted with methanol to disrupt the micelles, and spectrophotometrically analyzed at λ_max_ = 340 nm using an UV-Vis spectrophotometer (HP 8453, Hewlett Packard, Palo Alto, CA, USA). ATRA concentrations were determined in triplicate and were reported in a graph as means ± standard deviation (SD) versus concentrations of TPGS. TPGS solutions were used as blanks. The total ATRA content in the dispersions was evaluated on the basis of a calibration curve obtained by measuring the absorbance of standard solutions of ATRA in the same solvent. Data were acquired in triplicate and the means of three independent experiments ± SD were used to build the calibration graph.

### 3.3. Preparation of ATRA-TPGS Micelles

ATRA micelles were prepared by solvent-casting method [34]. In this method, a fixed amount of ATRA (6 mg) was solubilized with increasing amounts of TPGS (1:10, 1:20, 1:30, 1:40, 1:50, 1:60, and 1:70; *w*/*w*) in 6 mL of ethanol. The drug-to-polymer ratio was varied in order to determine the best encapsulation conditions. Then, the solvent was evaporated under vacuum at 40 °C by rotavapor and the melted mixture was cooled to room temperature to obtain a drug dispersed into a TPGS thin-layer film. Subsequently, the dried film was resuspended in an orbital shaker at room temperature for 4 h using 10 mL of Milli-Q water. Finally, the samples were filtered through a 0.22 µm filter to remove the excess non-encapsulated drug and used for further characterizations. Control formulations were prepared without ATRA, as described above. The micellar dispersions were also lyophilized using a freeze–dry system (Labconco, Kansas City, MI, USA). The freshly prepared micelles in water were first frozen at −20 °C, then placed into the lyophilization chamber set at −30 °C, and then, after the thermal equilibration of the matrix with the chamber temperature, indicated by a thermal probe inserted and congealed within a solution of a control sample, sublimation was performed by reducing pressure below 20 × 10^−3^ mbar. The primary drying lasted 48 h, then the secondary drying was carried out at 25 °C for at least 1 h. The amount of ATRA in the micelles was spectrophotometrically measured after dilution in methanol, as previously described, and the drug entrapment efficiency percentage (*EE*%) was subsequently calculated according to the formula in Equation (6).
(6)$$EE\%=\frac{\mathrm{Drug}\;\mathrm{wt}\;\mathrm{in}\;\mathrm{the}\;\mathrm{filtered}\;\mathrm{micellar}\;\mathrm{colloidal}\;\mathrm{dispersion}}{\mathrm{Drug}\;\mathrm{wt}\;\mathrm{added}\;\mathrm{in}\;\mathrm{the}\;\mathrm{preparative}\;\mathrm{mixture}}\;\times \;100$$

Differently, the drug loading capacity (*DL*%) was determined according to the formula in Equation (7), starting from a weighed lyophilized ATRA-TPGS powder sample reduced by TPGS CMC to consider only the fraction of TPGS forming the micelles, and assuming in both equations that the amount of free drug in the solution was negligible owing to its poor solubility in water.
(7)$$DL\%=\frac{\mathrm{wt}\;\mathrm{of}\;\mathrm{entrapped}\;\mathrm{drug}}{\mathrm{wt}\;\mathrm{of}\;\mathrm{the}\;\mathrm{loaded}\;\mathrm{micelles}-\mathrm{CMC}}\;\times 100$$

### 3.4. Determination of Micelles Size, Polydispersity Index (PDI), and Zeta (ƺ) Potential

The average micelle size Z_ave_ (mean hydrodynamic radius) and PDI were determined by PCS (photon correlation spectroscopy) using a Zetasizer 3000 HS instrument (Malvern Panalytical Ltd., Enigma Business Park, Grovewood Road, Malvern, WR14 1XZ, United Kingdom). Determinations were carried out at 25 °C at a fixed angle of 90°. Size analysis was performed on both fresh dispersions and on rehydrated freeze-dried powders. No additive was added during the liophylization process and no filtration was performed before the particle size analysis on the rehydrated freeze-dried powders [58]. All values were obtained after three runs of 10 measurements. The ƺ potential was measured through electrophoretic mobility by laser Doppler micro-electrophoresis. Three runs were made for each sample, and the results were reported as the mean of three independent measurements ± SD.

### 3.5. Principal Component Analysis (PCA)-Assisted FTIR Analysis

FTIR spectra were acquired on a PerkinElmer System 2000 spectrophotometer (PerkinElmer, Inc., Waltham, MA, USA) interfaced to a personal computer, and operating under Turbo chrome workstation (version 6.1.1., PerkinElmer, Inc., Waltham, MA, USA), Voltage: 120/240V. FTIR analysis of ATRA, TPGS, ATRA-TPGSs at 1:20, 1:30, 1:40, 1:50, 1:60 and 1:70 *w*/*w*, and additionally more concentrated ATRA at 1:1, 1:5, and 1:10 *w*/*w*, were performed by formulating the samples in KBr pellets.

#### 3.5.1. FTIR Spectra Acquisition

The FTIR spectra were acquired in triplicates for each compound in transmission mode, and the matrix of spectral data of all the spectra acquired were subjected to principal component analysis (PCA) using R statistical software, free downloadable (http://cran.mirror.garr.it/mirrors/CRAN/ (Accessed on 3 March 2021) Garr Mirror, Milan, Italy). All spectra were recorded both in the transmission and absorption modes, from 4000 to 800 cm^−1^, with 1 cm^−1^ spectral resolution, co-adding 32 interferograms, and with a measurement accuracy in the frequency data at each measured point of 0.01 cm^−1^ due to the laser internal reference of the instrument. The frequency of each band was obtained automatically by using the ‘‘find peaks’’ command of the instrument software.

#### 3.5.2. Chemometric Analysis: PCA

All the FTIR data sets, each obtained from triplicate acquisition, were processed in *n* measurable variables. For each sample, the variables consisted of the values of absorbance (%) associated to the wavenumbers (3001) in the range 4000−1000 cm^−1^. In order to simplify the system, we exploited the PCA, which reduced the large number of variables to a small number of new variables, namely, the principal components (PCs). Chemometric analyses by PCA were performed on a matrix of data of 11 × 3001 including 33011 variables, and by collecting FTIR data of the 11 samples considered in the study. Spectral data were preprocessed by the standard normal variate (SNV) transformations to minimize the global intensity effects due to slightly different optical paths and column mean-centering.

### 3.6. Differential Scanning Calorimetry (DSC)

To confirm the entrapment of ATRA inside the micelles, DSC analysis was performed. The thermal properties of lyophilized ATRA-TPGSs, free ATRA and TPGS, and of physical mixtures of the raw materials in equal ratios to those in the prepared ATRA-TPGSs were studied using DSC-7 equipped with Pyris software (PerkinElmer, Inc.,Waltham, MA, USA). The instrument was calibrated with Indium and Zinc and about 4 mg of samples were crimped in aluminum pans. The thermograms were recorded from 10 to 250 °C at a heating rate of 10 °C/min under nitrogen flow.

### 3.7. Stability Studies

To investigate if the initial drug loading makes a difference in the thermodynamic stability of the micelles prepared by solvent casting, a stability study was performed on freshly prepared micelle dispersions. The samples were maintained at 25 °C in the incubator WTB © BINDER GmbH 2015–2020 (Im Mittleren Ösch 5, D-78532 Tuttlingen, Germany) and observed after 24, 48, and 72 h. This study consisted of visual inspections for signs of precipitation or crystal growth, time-dependent changes in mean diameters, and spectrophotometrical detection of drug concentrations in the solution after filtration through a 0.22 μm syringe filter to remove ATRA leakage. From each filtered solution, 1.0 mL was subsequently withdrawn, then diluted to 1:50 with methanol, and analyzed on the spectrophotometer. The determinations were made in triplicate and results were reported as mean ± SD.

### 3.8. Incorporation of Micelles in Hydrogels

Three different concentrations (0.5%, 1.0%, and 1.5% *w*/*w*) of Carbopol^®^ 980 were used to prepare the gel base. The required amount of the gelling agent was added to an exactly weighed aqueous micellar dispersion and let soak overnight; then, the hydrated polymer suspensions were neutralized by the addition of triethanolamine (0.2% *w*/*w*) to obtain an appropriate consistency and a suitable pH. Hydrogels prepared with void micelles were used as the blank. All formulations were prepared in triplicate and stored until use at room temperature, protected from light. Moreover, the pH was determined by a calibrated pH-meter after the dilution of the obtained gel at 10% (*w*/*w*) in milli-Q water. Finally, the hydrogels were checked for ATRA crystals through light microscopy.

### 3.9. Rheological Studies

For the rheological characterization of the gels, a concentric cylinder viscometer (Phisica Haake Thermo Fisher Scientific Inc., Waltham, MA, USA) equipped with a Z4 probe was used. For the determinations of dynamic viscosity, approximately 5 g of hydrogel was placed in the viscometer and thermally equilibrated at the test temperature (25 ± 0.1 °C) for 60 min before the measurement; then, the sample was submitted to an increasing shear rate from 0 to 500 s^−1^ by a 100 s^−1^ /min gradient. All the experiments were performed in triplicate and mean values were used in analysis.

### 3.10. Skin Permeation Experiments

The studies were performed using a 7.5 mL static Franz diffusion of vertical cells with a diameter of 10 mm through excised pig ear skin. Newborn pig skin was obtained from a local slaughterhouse and stored at −20 °C until use. The subcutaneous tissue and fat were carefully removed from the dermal surface and circular pieces were sandwiched tightly between the two compartments after equilibration in the phosphate buffer (PBS, 1 mM, pH 7.4) at 32 ± 1 °C for 2 h. The receptor compartment was filled with 7.5 mL of ethanol:PBS solution (50:50 *v*/*v*) to maintain sink conditions, and stirred continuously with a magnetic bar (100 rpm) [10]. The ATRA hydroalcoholic solution and the formulations obtained using ATRA:TPGS at 1:50, both as micellar dispersion and as hydrophilic gel with 1.0% of Carbopol^®^ 980, were applied on the stratum corneum at an initial ATRA concentration of near 0.3 mg/mL. Samples without ATRA were used as blanks. The experiments were carried out in open conditions for 24 h. The determination of ATRA concentration in the receptor liquid was made spectrophotometrically (λ_max_ = 340 nm) using an UV-Vis spectrophotometer and the calibration curve with Equation (1), made as described in Section 3.2 [59]. The temperature was maintained at 32 ± 1 °C, which is equal to the stratum corneum physiologic temperature. The experiments were performed by avoiding light exposure. Determinations were made in triplicate and were expressed as mean ± SD.

### 3.11. Cell Viability Studies

The primary human BRAFV600 melanoma cell line was supplied by Dr. Gabriella Pietra (DIMES, University of Genoa, Genoa, Italy). Cells were periodically tested for mycoplasma contamination (Mycoplasma Reagent Set, Aurogene s.p.a, Pavia, Italy) and were cultured in a RPMI 1640 medium (Euroclone, Milan, Italy) supplemented with 10% fetal bovine serum (Euroclone), 2 mM glutamine (Sigma-Aldrich, Milan, Italy), and 1% penicillin/streptomycin (Sigma-Aldrich), and maintained in a 5% CO_2_ humid atmosphere. Cells were seeded (15 × 10^3^ cells/well) in 96-well plates (Corning Incorporated, New York, NY, USA) and then exposed for 72 h to free ATRA in a range of concentrations between 0.1 and 20 µM, in order to determine the toxic concentration. The drug was solubilized in DMSO and then diluted with a culture medium to a maximum final solvent concentration of 0.075%. In the experiments with ATRA-TPGSs, cells were exposed for 72 h to ATRA (5 and 7.5 µM), ATRA-TPGSs at 1:50 (in concentrations able of providing 5.0 and 7.5 µM of ATRA), and TPGS (at the same concentrations provided by the amounts of ATRA-TPGSs tested). Cell viability was determined by using the CellTiter 96^®^ Aqueous One Solution Cell Proliferation Assay (Promega, Madison, WI, USA), as previously reported [60].

### 3.12. Statistical Analysis

All the experiments were performed at least in triplicate. The results from cell viability studies were expressed as mean ± SD (standard deviation). A one-way analysis of variance (ANOVA) was employed for the comparison of the experimental data. Differences were considered significant at a level of *p* < 0.05 or *p* < 0.01. Concerning other results, the significance of the differences between the values was assessed using the *t*-test performed using the paleontological statistics software package for education and data analysis PAST, freely downloadable online at: https://past.en.lo4d.com/windows (accessed on 3 March 2021). Differences were considered statistically significant when *p* < 0.05. In the tables and figures, all data are reported as mean value ± SD and statistical differences are signaled with *.

## 4. Conclusions

The list of diseases that may be treated by topical administration of ATRA is remarkably large, including, among others, acne, actinic keratoses, psoriasis, and melanoma. Based on these preliminary results, we demonstrated the feasibility to prepare aqueous semisolid formulations containing ATRA with suitable characteristics for dermatological administration, avoiding the use of organic solvents. As a solubilizing agent, we chose TPGS, a FDA-approved excipient, for its important feature to be a safe pharmaceutical adjuvant. First, we prepared polymeric micelles of nanometric dimensions able to encapsulate ATRA at desirable concentrations, and then, we included them into a hydrogel. Up to now, the formulation of a TPGS-based nanocarrier-loaded gel had not been investigated. Interestingly, the ATRA polymeric micelles showed a significantly improved antiproliferative activity against melanoma cells in comparison to free ATRA. Additionally, TPGS alone exerted significant cytotoxic activity versus the control cells, suggesting the presence of an additive effect of the two substances. This enhanced activity could open a new insight in the treatment of tumors responding to ATRA, like melanoma. Indeed, ATRA-TPGSs are toxic for melanoma cells at concentrations that do not affect normal skin fibroblasts. Studies regarding the shelf-life of frozen aqueous dispersions and of the lyophilized powders maintained at 4 °C are currently in progress and have demonstrated good storage stability thus far. Moreover, in vitro biological studies on neuroblastoma and melanoma cells lines will be performed in order to elucidate the cellular pathways and the combined effect of ATRA and TPGS.

## Figures and Tables

**Figure 1 pharmaceuticals-14-00212-f001:**
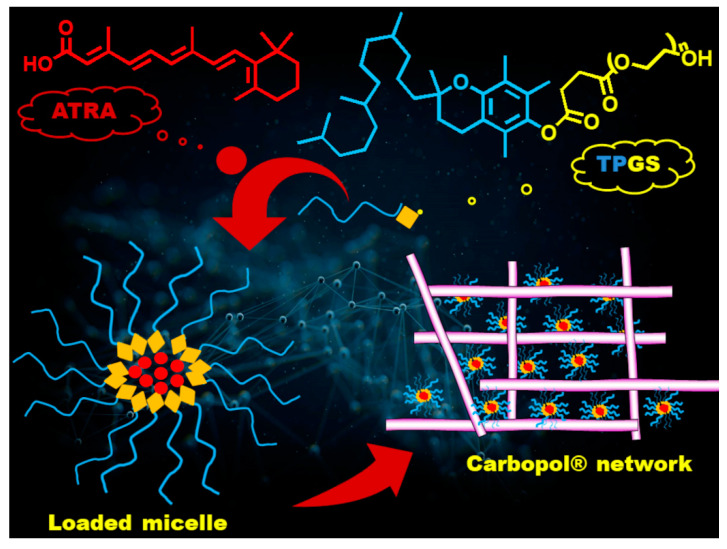
Chemical structures and schematic representation of micelles and their dispersion into the Carbopol^®^ network.

**Figure 2 pharmaceuticals-14-00212-f002:**
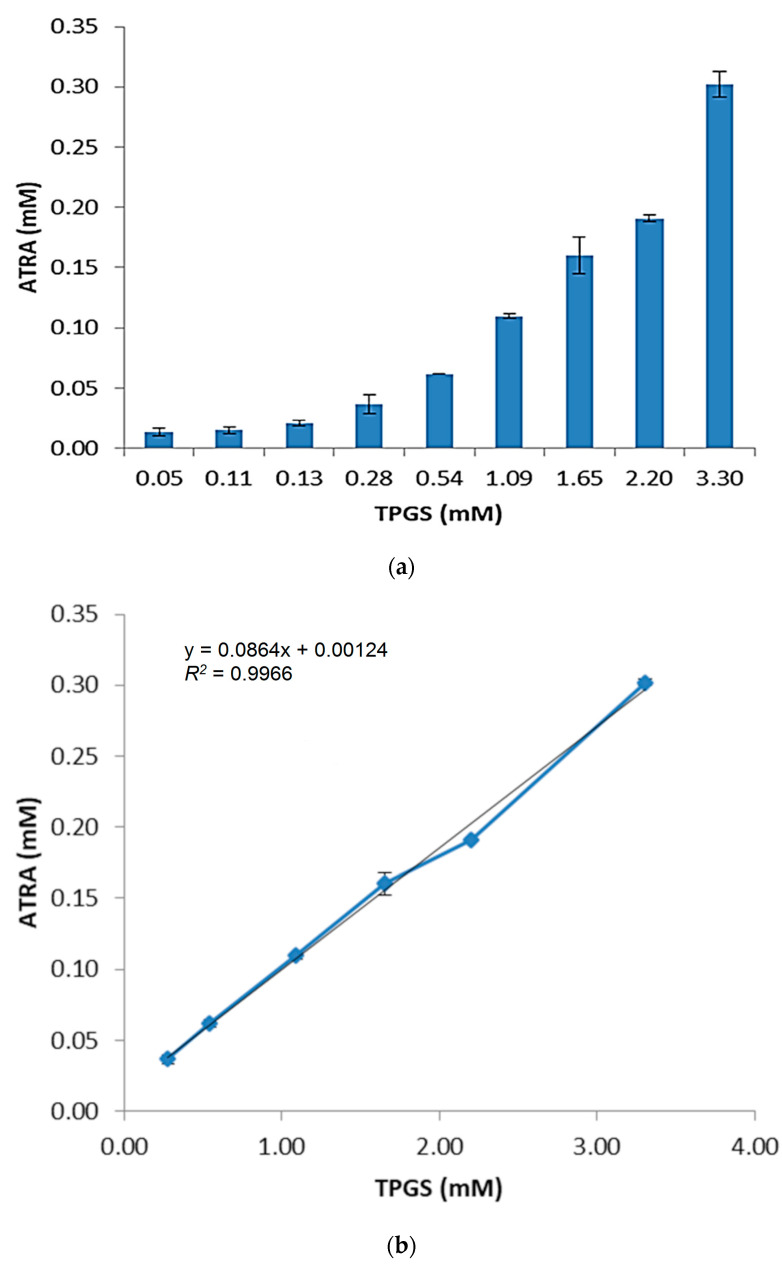
D-α-tocopheryl-polyethylene-glycol-succinate (TPGS) concentration-dependent solubility of all-*trans*-retinoic acid (ATRA). Total solubility of ATRA versus TPGS concentration measured by equilibrium method after 48 h at 37 °C in water (**a**). Linear regression obtained considering the values above TPGS critical micellar concentration (CMC) (**b**).

**Figure 3 pharmaceuticals-14-00212-f003:**
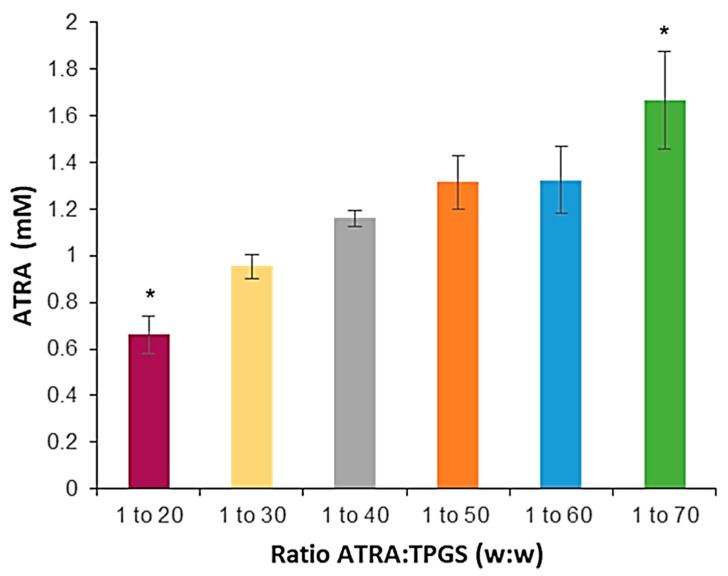
ATRA total solubility obtained by solvent-casting method as a function of ATRA:TPGS *w*/*w* ratio present in the starting mixture. Histogram summarizes quantitative data of the means ± S.D. of three independent experiments. * indicates statistical difference vs. sample mean (*p* < 0.05).

**Figure 4 pharmaceuticals-14-00212-f004:**
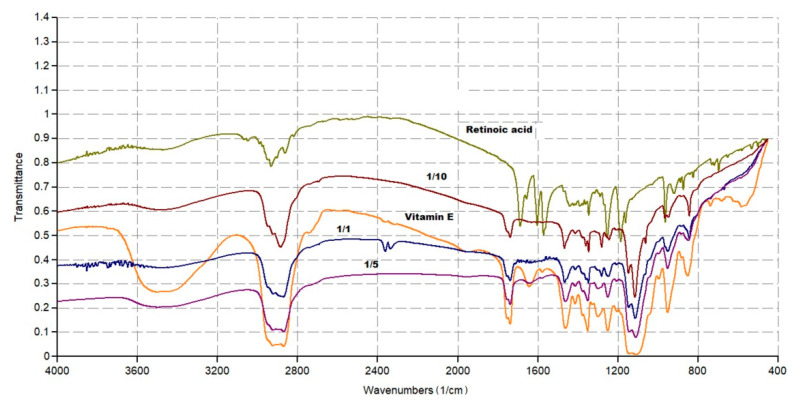
Fourier transform infrared spectroscopy (FTIR) spectra of ATRA (retinoic acid), TPGS (vitamin E) and of three ATRA-TPGS formulations.

**Figure 5 pharmaceuticals-14-00212-f005:**
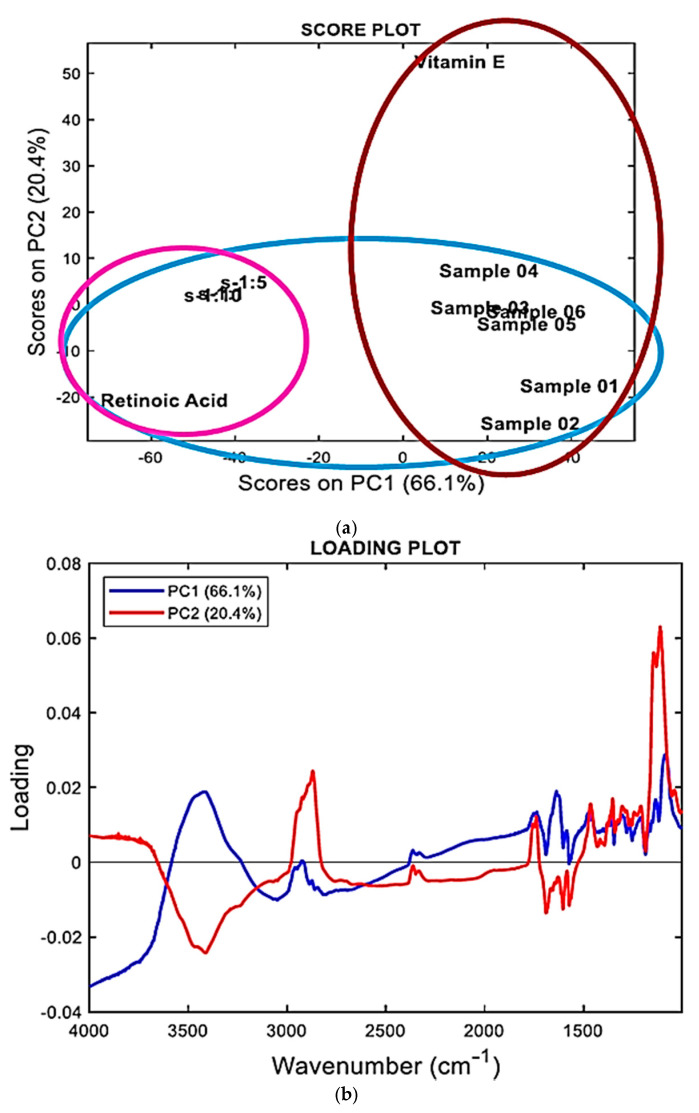
Principle component analysis (PCA) results represented as a score plot (**a**) and a loading plot (**b**), performed on the matrix by collecting spectral data of all samples using R software.

**Figure 6 pharmaceuticals-14-00212-f006:**
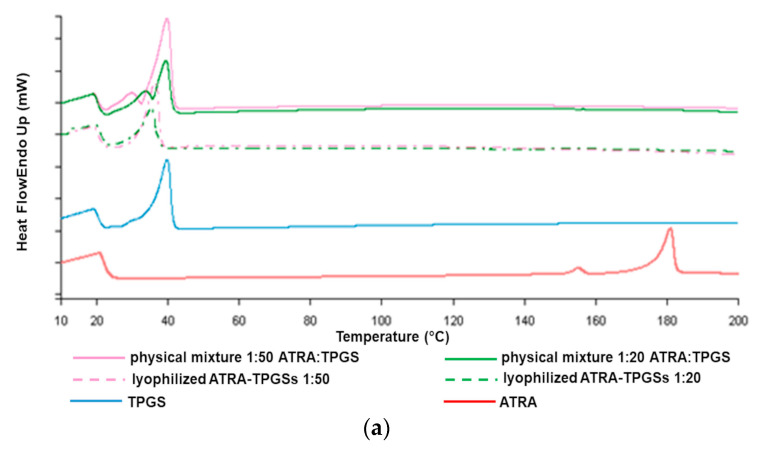
Differential scanning calorimetry (DSC) thermograms of selected ATRA-TPGS formulations, ATRA:TPGS physical mixtures, TPGS, and ATRA (**a**). Overlapped thermograms with y axis reporting the real scale (**b**).

**Figure 7 pharmaceuticals-14-00212-f007:**
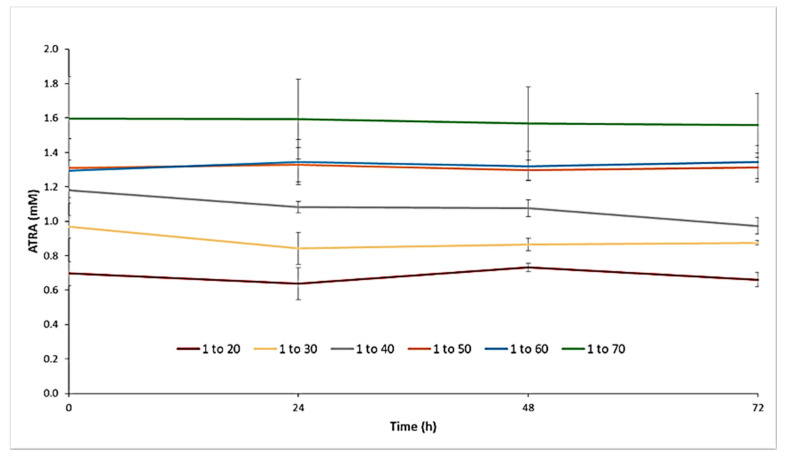
Stability over time of aqueous micellar dispersions maintained at 25 °C and obtained using ATRA:TPGS 1:20, 1:30, 1:40, 1:50, 1:60, and 1:70 (*w*/*w*) ratios in the preparative mixture.

**Figure 8 pharmaceuticals-14-00212-f008:**
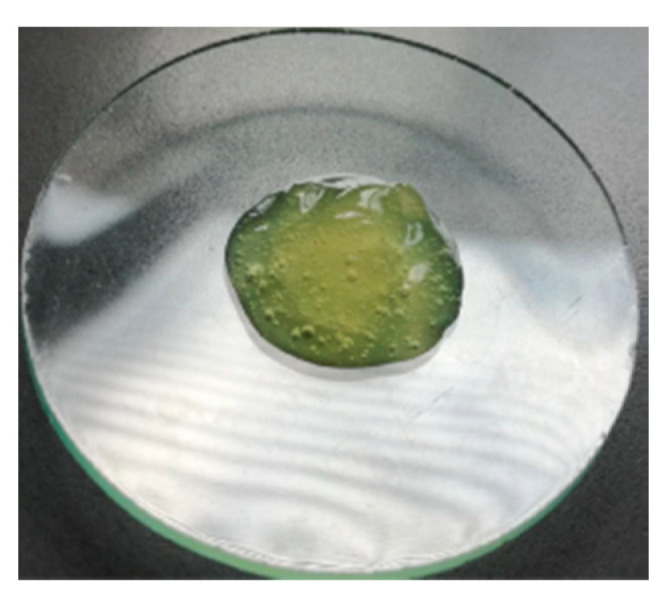
Carbopol ^®^ 980 gel loaded with the colloidal dispersion of ATRA-TPGSs prepared from ATRA:TPGS 1:50 (*w*/*w*) ratio.

**Figure 9 pharmaceuticals-14-00212-f009:**
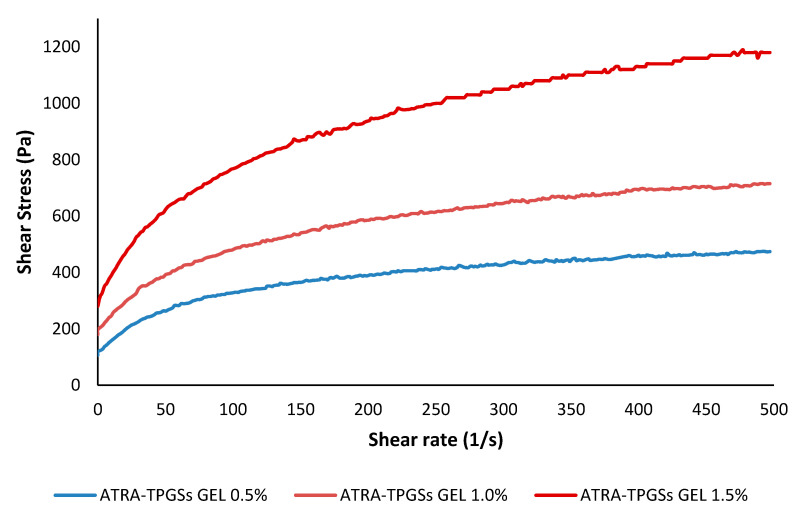
Rheograms of ATRA-TPGS-loaded hydrogels at 0.5%, 1.0%, and 1.5% (*w*/*w*) Carbopol^®^ 980 concentrations, recorded at 25 °C.

**Figure 10 pharmaceuticals-14-00212-f010:**
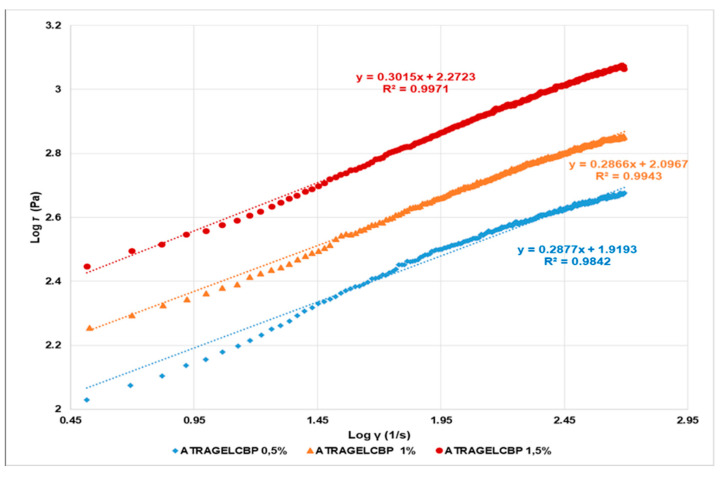
Power Law rheograms of ATRA-TPGS-loaded gels at 0.5%, 1.0%, and 1.5% (*w*/*w*) Carbopol^®^ 980 concentrations.

**Figure 11 pharmaceuticals-14-00212-f011:**
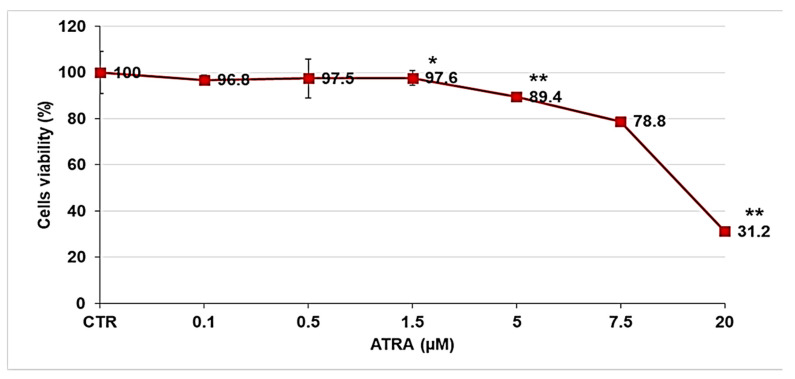
Dose-dependent cytotoxic activity of ATRA on melanoma cells. The analysis was performed by CellTiter 96^®^ Aqueous One Solution Cell Proliferation Assay in melanoma cells exposed to ATRA (0.1–20 µM) for 72 h. The graph summarizes the quantitative data of the means ± S.E.M. of three independent experiments. ** *p* < 0.01 vs. Ctr cells; * *p* < 0.05 vs. Ctr cells.

**Figure 12 pharmaceuticals-14-00212-f012:**
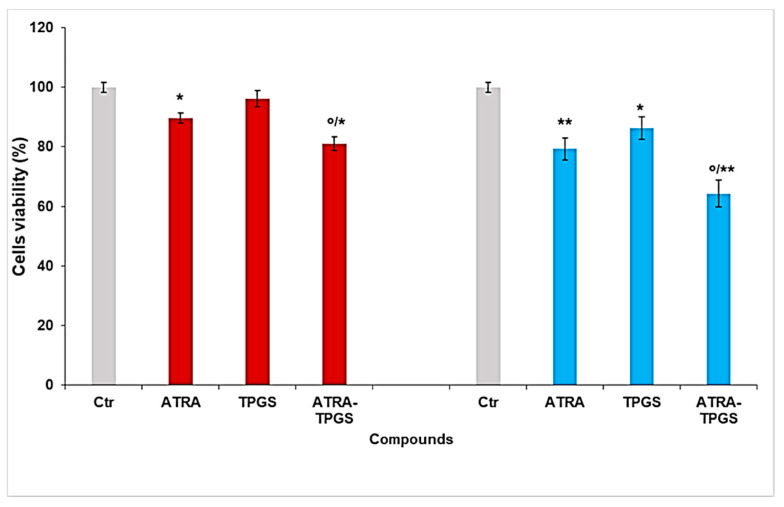
Effect of ATRA-TPGSs on melanoma cell viability. Cell viability was evaluated by CellTiter 96^®^ Aqueous One Solution Cell Proliferation Assay. Cells were exposed for 72 h to free and encapsulated ATRA 5 µM (red columns) and 7.5 µM (light blue columns), and to TPGS at the same concentrations provided by the amounts of ATRA-TPGSs. Histograms summarize the quantitative data of the means ± S.E.M. of three independent experiments. * *p* < 0.05 vs. Ctr cells; ** *p* < 0.01 vs. Ctr cells; ° *p* < 0.05 vs. ATRA-treated cells.

**Table 1 pharmaceuticals-14-00212-t001:** Encapsulation efficiency (EE%) and drug loading capacity (DL%) of different ATRA-TPGS formulations. Each value represents the mean ± standard deviation (± SD) (degrees of freedom = 3).

ATRA:TPGS (*w*/*w*)	EE%	DL%
1:20	35 ± 2 *	0.76 ± 0.02 *
1:30	47 ± 3	0.86 ± 0.03
1:40	58.6 ± 0.2	0.88 ± 0.01 *
1:50	65 ± 3	0.83 ± 0.03
1:60	68 ± 2	0.78 ± 0.02
1:70	79 ± 8 *	0.79 ± 0.05

* indicates statistical difference vs. sample mean (*p* < 0.05).

**Table 2 pharmaceuticals-14-00212-t002:** Mean diameter (size, nm) and polydispersity index (PDI) of freshly prepared ATRA-TPGS formulations reconstituted from lyophilized powder. The zeta potential (ƺ) values were detected on fresh ATRA-TPGS micelles in water and in the HEPES buffer. Each value represents the mean ± SD (degrees of freedom = 3).

ATRA:TPGS (*w*/*w*)	Size (nm) Fresh	PDI Fresh	Size (nm) Lyophilized	PDI Lyophilized	ƺ (mV) mQ Water	ƺ (mV) HEPES
1:20	13.8 ± 0.1	0.24 ± 0.03	15.0 ± 0.6	0.32 ± 0.09	−7.2 ± 0.5 *	0 ± 3 *
1:30	17 ± 5 *	0.3 ± 0.1 *	19 ± 5	0.31 ± 0.02	−9 ± 3	5 ± 1
1:40	12.6 ± 0.4	0.17 ± 0.06	14.1 ± 0.8	0.2 ± 0.1	−13 ± 2 *	0 ± 2 *
1:50	11.9 ± 0.4	0.13 ± 0.06	14.7 ± 0.1	0.212 ± 0.001	−11 ± 1	5 ± 1
1:60	11.8 ± 0.4	0.14 ± 0.03	21 ± 7 *	0.4 ± 0.1 *	−13 ± 3 *	4 ± 2
1:70	11.4 ± 0.1	0.11 ± 0.01	14.4 ± 0.2	0.19 ± 0.03	−10.1 ± 0.7	4 ± 1

* indicates statistical difference vs. sample mean (*p* < 0.05).

**Table 3 pharmaceuticals-14-00212-t003:** Equations, correlation coefficients (*R*^2^), and slope and intercept values of the linear regressions obtained by fitting the rheograms data to the mathematical models.

Model	Equations	*R* ^2^	Slope	Intercept
Power Law	*y* = 0.2877*x* + 1.9193 ^1^ *y* = 0.2866*x* + 2.0967 ^2^ *y* = 0.3015*x* + 2.2723 ^3^	0.9842 0.9943 0.9971	0.2877 0.2866 0.3015	1.9193 2.0967 2.2723
Herschel–Bulkley	*y* = 0.5527*x* + 1.1508 ^1^ *y* = 0.5899*x* + 1.2154 ^2^ *y* = 0.6090*x* + 1.3886^3^	0.8139 0.8308 0.8226	0.5527 0.5899 0.6090	1.1508 1.2154 1.3886
Casson	*y* = 0.4389*x* + 12.857 ^1^ *y* = 0.5328*x* + 15.915 ^2^ *y* = 0.7189*x* + 19.668 ^3^	0.8980 0.9351 0.9491	0.4389 0.5328 0.7189	12.857 15.925 19.668
Bingham Plastic	*y* = 0.5178*x* + 257.98 ^1^ *y* = 0.8055*x* + 378.43 ^2^ *y* = 1.3961*x* + 593.18 ^3^	0.8350 0.8650 0.8873	0.5178 0.8055 1.3961	257.98 378.43 593.18

ATRA-TPGS-loaded gel with Carbopol ^1^ 0.5%, ^2^ 1.0%, and ^3^ 1.5% (*w*/*w*).

**Table 4 pharmaceuticals-14-00212-t004:** Comparison of cumulative ATRA amount permeated from different formulations through porcine skin after 24 h. Data represent the mean ± SD of at least four experimental determinations.

Formulation	Amount Permeated (µg cm^−2^ ± SD)
ATRA-TPGS dispersion	27 ± 5
ATRA-TPGS-loaded gel	22 ± 4
ATRA hydroalcholic solution	32 ± 3

## Data Availability

The data presented in this study are available on request from the corresponding author.

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
