# Peer review of "D-α-Tocopherol-Based Micelles for Successful Encapsulation of Retinoic Acid"

_pharmaceuticals, 2021, doi:10.3390/ph14030212_

Round 1

Reviewer 1 Report

Zuccari and coworkers investigate TPGS micelles for encapsulation of retinoic acid. The results are interesting, and the characterizations are sufficient. However, it could be published in Pharmaceuticals only the following issue are addressed.

  1. For the readers, it will be better if the authors could add one figure schematically shown how ATRA is encapsulated into TPGS micelles, including the chemical structures of ATRA, TPGS, and Carbopol® 980 (carbomer).
  2. Line 21, ‘Z’ should be changed to ‘zeta’.
  3. Line 61-63, ATRA could be also loaded when it is conjugated to a polymer. One recent study (doi.org/10.1021/acsnano.9b04166) should be included regarding to this point.
  4. In figure, the percentage of DMSO used for high concentration of ATRA needs to be noted in the methods.
  5. The resolution of figure 1, 2, 6, 9, 10, and 11 is not high enough.
  6. Figure S3, the size of the micelle is quite small. However, it is difficult to get one peak with intensity-based distribution. Please give the size distribution result with wider range but not only from 0 to 20 nm.
  7. Drug loading capacity is the amount of drug loaded per unit weight of the nanoparticle (including the carrier and the drug). However, from equation 7, the authors used mole instead of the weight, and the denominator doesn’t contain the drug ATAR but only the TPGS. This makes it hard to compare with the result in the literature.

Author Response

  1. For the readers, it will be better if the authors could add one figure schematically shown how ATRA is encapsulated into TPGS micelles, including the chemical structures of ATRA, TPGS, and Carbopol® 980 (carbomer).

The Authors agree with the reviewer and a new figure (Figure 1, under line 155) has been added with a sketch of chemical structures and of the micelle-loaded gel.

  1. Line 21, ‘Z’ should be changed to ‘zeta’.

The correction has been made (line 21).

  1. Line 61-63, ATRA could be also loaded when it is conjugated to a polymer. One recent study (doi.org/10.1021/acsnano.9b04166) should be included regarding to this point.

As explained in the sentence (line 84), the references cited by us in the manuscript are all referred to ATRA dermal delivery, while the reference proposed by the Reviewer concerns a study regarding the intravenous ATRA administration. Therefore, in our opinion the paper is out of the field of interest.

  1. In figure, the percentage of DMSO used for high concentration of ATRA needs to be noted in the methods.

The DMSO percentage used for ATRA has been added in the methods (line 877).

  1. The resolution of figure 1, 2, 6, 9, 10, and 11 is not high enough.

With regard to the resolution of the Figures mentioned by the Reviewer, we would kindly point out that the original Figures provided to Pharmaceuticals in a separate folder were obtained with a resolution of 700 dpi, i.e. far higher than that requested by the journal. However, to satisfy the Reviewer request, all the Figures inserted in the manuscript have been furtherly improved, by playing on the contrast, luminosity and size.

  1. Figure S3, the size of the micelle is quite small. However, it is difficult to get one peak with intensity-based distribution. Please give the size distribution result with wider range but not only from 0 to 20 nm.

Actually, a Figure with a wider range (i.e. 0-100 nm) was already provided in the original SI (Figure S4). In this Figure, a representative size distribution of a lyophilized and fresh formulation is reported with a range from 0 to 100 nm and, since the lyophilization process may induce aggregation, we reported only here a wider rage to demonstrate the absence of aggregates.  Anyway, to satisfy the Reviewer request,  we have included an additional new Figure (Figure S3b, SI) reporting the full scale. As you can see it  do not allow to discriminate every single formulation, therefore we preferred showing a smaller range.

  1. Drug loading capacity is the amount of drug loaded per unit weight of the nanoparticle (including the carrier and the drug). However, from equation 7, the authors used mole instead of the weight, and the denominator doesn’t contain the drug ATAR but only the TPGS. This makes it hard to compare with the result in the literature.

The Reviewer is right, then the Drug Loading formula has been changed and the DL% values reported per unit of weight (line 767).

Reviewer 2 Report

Comments and Suggestions for Authors

 Micelles consisting of TPGS is not new but micelles integrated hydrogels are interesting as it is new concept. However, the paper lacks critical information as mentioned in below to scientifically support the usefulness of the new formulations.

  1. The introduction part is poorly written and thus, it is difficult to find the significance and novelty of the paper. What were the ultimate goals of current formulations?

  1. Scientific background of the formulation of Retinoic acid with nanocarriers is not clear. The authors need to analyze existing technologies and formulations of Retinoic acid to identify the advantages of current approaches.
  2. The reason of incorporation of micelles in hydrogel? Author need to elaborate the rationale behind it.
  3. What amount of micelles was incorporated in hydrogels? And what was quantity of drug loaded in micelles?
  4. Does author measure the influence of triethanolamine on micelles stability?
  5. Why cell viability study was performed with micelles only? Not with finished product?
  6. Does author confirm the crystal increase of drugs through XRD studies?
  7. Did author confirm the reformation of micelles after incorporation in hydrogel? Did micelles remain intact?
  8. What was the selection criteria of micelles formulation for incorporation in hydrogel?
  9. Across the data sets, statistical analysis is missing (e.g., Ex vivo permeation study). Please conduct statistical analysis to make meaningful data comparisons.
  10. Conclusion is not related to claim of study. Is TPGS complexation only reason of anti-proliferative activity?
  11. Authors need to improve the design of study by developing relationship between drug, micelles, hydrogel and cytotoxic effects.  

Author Response

  1. The introduction part is poorly written and thus, it is difficult to find the significance and novelty of the paper. What were the ultimate goals of current formulations?

In the introduction a deeper description of previous studies concerning ATRA topical administration has been added, highlighting the main features of each one. Moreover, the novelty of the paper has been explained in a more clearly way. Please, see all the parts added to satisfy the requests.

  1. Scientific background of the formulation of Retinoic acid with nanocarriers is not clear. The authors need to analyze existing technologies and formulations of Retinoic acid to identify the advantages of current approaches.

We agree with the Reviewer, therefore a description of current formulations present in the market and the advantages associated to the nanotechnological approaches were included in the introduction.

  1. The reason of incorporation of micelles in hydrogel? Author need to elaborate the rationale behind it.

The explanation of the incorporation of the micelles in a gel has been added in the subsection 2.6. from lines 491 to line 495.

  1. What amount of micelles was incorporated in hydrogels? And what was quantity of drug loaded in micelles?

These data were already provided in the unrevised version of the manuscript in the subsection 2.7. Please, see line 602 and line 613.

  1. Does author measure the influence of triethanolamine on micelles stability?

We have only measured the final pH of the gel that was in the range from 5.0 to 5.8, because, according to the manufacturer data sheet, TPGS is stable under these conditions. As a confirmation, please see the following link:

https://pmcisochem.fr/sites/default/files/documents/TPGS%20ISODEL%20-%20June%202020.pdf.

  1. Why cell viability study was performed with micelles only? Not with finished product?

Because in case of an in vivo topical administration, Carbopol will act simply as a vehicle and will not penetrate into the skin and therefore into the cells, due to its high molecular weight. It will remain at the stratum corneum level, so in vivo, cells viability will be affected only by drug loaded micelles. To make experiments with the finished product would be nonsense.

  1. Does author confirm the crystal increase of drugs through XRD studies?

Unfortunately, we do not possess the required instrument. However, we made a visual inspection of the samples and then, we determined the loss of drug in solution by spectrophotometric analysis after filtration of the dispersions as many other previous studies suggested. Please see at Abouzeid AH, Patel NR, Torchilin VP.Int J Pharm. 2014 Apr 10;464(1-2):178-84. doi: 10.1016/j.ijpharm.2014.01.009. Accordingly, we kindly ask the auditor to be satisfied.

  1. Did author confirm the reformation of micelles after incorporation in hydrogel? Did micelles remain intact?

The micelles remained intact after incorporation into the gel base. This was confirmed observing the gel under the microscope, to detect drug eventually precipitated and by PCS analysis after gel dilution in water. To a better comprehension for the readers, we have added this information at lines 510-512.

  1. What was the selection criteria of micelles formulation for incorporation in hydrogel?

We have explained the reasons at lines 475-484.

  1. Across the data sets, statistical analysis is missing (e.g., Ex vivo permeation study). Please conduct statistical analysis to make meaningful data comparisons.

As requested, statistical analyses have been performed for all experiments where necessary by executing the t-test or the one-way ANOVA using a the free downloadable statistical software reported in the main text.

  1. Conclusion is not related to claim of study. Is TPGS complexation only reason of anti-proliferative activity?

We agree with the Reviewer. Indeed, we had reported that also TPGS alone has a partial effect on the viability of melanoma cells suggesting an additive effect of the two substances only in the main text, but we had not mentioned this property in the conclusions. Consequently, we have added this aspect (lines 909-914). In addition, we have also changed the first part of the conclusion, according to the Reviewer suggestions. 

  1. Authors need to improve the design of study by developing relationship between drug, micelles, hydrogel and cytotoxic effects.  

As suggested, we have harmonized the text, explaining with more precision the steps and the connections among the various experiments. The discussion has been also expanded in an attempt to provide a more in-depth description and discussion of the results obtained. We thank the Reviewer for his suggestions.

Reviewer 3 Report

The manuscript by Zuccari et al. describes a rigorous study on formulation and ex vivo evaluation of a gellified form of complexed ATRA. The study is well conducted and claims are effectively supported by experimental evidences. I really enjoyed reading it.

I have only a couple of minor questions to be addressed before the manuscript is publishable.

1) rows 186 and below: please revise digits to the first error digit (i.e. 79 +- 8 and not 79.1 +- 8.3 and so on. The last significant digit of the measure should be the same of the first digit of the error). This should be checked in thorough the entire manuscript.

2) table 2: PDI increases, as correctly stated by the authors, after freeze drying. i wonder whether this is due to a simple broadening or to the appearance of a very minor fraction of large aggregates. In this last case, the fact should be evidenced, because the actual broadening in the micelles of interest could actually be significantly better than what shown.

3) rows 315 and following: I think that a long explanation of how to interpret PCA graphs is unnecessary given the target audience of this manuscript. Please short this part.

Author Response

The manuscript by Zuccari et al. describes a rigorous study on formulation and ex vivo evaluation of a gellified form of complexed ATRA. The study is well conducted and claims are effectively supported by experimental evidences. I really enjoyed reading it.

I have only a couple of minor questions to be addressed before the manuscript is publishable.

1) rows 186 and below: please revise digits to the first error digit (i.e. 79 +- 8 and not 79.1 +- 8.3 and so on. The last significant digit of the measure should be the same of the first digit of the error). This should be checked in thorough the entire manuscript.

We agree with the Reviewer, indeed the significant digitals are those certain, plus the first digit of uncertainty. Consequently, the whole manuscript was modified accordingly in both the main text and Tables.

2) table 2: PDI increases, as correctly stated by the authors, after freeze drying. i wonder whether this is due to a simple broadening or to the appearance of a very minor fraction of large aggregates. In this last case, the fact should be evidenced, because the actual broadening in the micelles of interest could actually be significantly better than what shown.

The phenomenon of the increased particle size and PDI following lyophilization has been extensively studied for both liposomes and micelles. Lyophilization represents a stressing process for particles and various substances have been tested as cryoprotectants, in order to limit these variations. As for the micelles, this increase is mainly due to the particle aggregation tendency during the removal of the solvent.

Please see: C. Di Tommaso et al. / European Journal of Pharmaceutical Sciences40 (2010) 38–47.

For more clarity, a sentence explaining these reasons, has been added in the main text (lines 294-295)

3) rows 315 and following: I think that a long explanation of how to interpret PCA graphs is unnecessary given the target audience of this manuscript. Please short this part.

We agree with the Reviewer and this section has been shortened.

Reviewer 4 Report

In this manuscript, the authors developed and evaluated TPGS loaded ATRA. Numerous studies have been performed to evaluate the best possible ratio of ATRA to TPGS as well as permeation and cytotoxicity studies are also included. However, overall I found a lack of novelty aspect in the manuscript and a poorly written discussion part. The advantage of this system compared to previously reported results are clearly missing. I do not recommend the publication of this manuscript in the current form.

Some specific concerns are provided below:  

  • The statement in lines 91-92 is contradicted. See the below articles:

https://www.tandfonline.com/doi/abs/10.3109/10837450.2013.763261?journalCode=iphd20

  • Line 123, I would not consider it a high value of correlation coefficient if it's below 0.995
  • Please check for spellings, intramuscular for example. Line 151.
  • Figure 2, please perform statistical analysis, most of the values presented are non-significant. Why there is no increment in solubility/encapsulation with increasing TPGS unlike in the solubility study explained right before.
  • The size data is not correlating with the TPGS quantity used. Why? Further, it really hard to follow the graphs provided in SI. Can you please include original graphs generated from zetasizer in full range?
  • Line 226, negative zeta potential was attributed to free drug but how much drug could be expected in water, which could influence the overall charge. It will be good to analyze free drug in water to prove this concept.
  • Table 4, indeed having PEG on the surface influence the permeation rate due to the obvious reason of hydrophilicity. However, on the other hand, nanosized drug carrier demonstrates higher permeation owing to the size. The permeation results are not impressive, than what’s the overall purpose of this study?
  • Figure 11, why TPGS-ATRA is more toxic? No discussion provided.

Author Response

Our responses have been attached as word file.

Round 2

Reviewer 1 Report

While the authors answered most of my comments, I still have doubts about the size distribution of the lyophilized sample. It is almost impossible to get a single  peak in the size distribution of the lyophilized sample (there will be strong aggregation in PEG stablized system during the rehydration). I would like to know whether there is additive added during the lyophilization process or a filtration step performed before the size measurement. Such information should be included in the Methods. If not, the authors should add the discussion in the text to explain why the system is so robust.

Author Response

While the authors answered most of my comments, I still have doubts about the size distribution of the lyophilized sample. It is almost impossible to get a single peak in the size distribution of the lyophilized sample (there will be strong aggregation in PEG stablized system during the rehydration). I would like to know whether there is additive added during the lyophilization process or a filtration step performed before the size measurement. Such information should be included in the Methods. If not, the authors should add the discussion in the text to explain why the system is so robust.

The authors do not understand the Reviewer doubt. As the Reviewer can observe in Figure S4 (SI), only one peak for the liophylized sample was detected in the range scale 0-100 nm, confirming that no additional dimensional family existed. Anyway, the tendency of particles to form some small aggregates, not detectable by the instrument, but influencing only minimally the particle mean size and PDI, that resulted slightly increased, has been notified and discussed by us ourselves. Anyway, to answer the Reviewer question and for more clarity, we have inserted in the Methods, that no filtration was performed before size analysis and no additive was added during liophylization procedure (759-761 of the pdf-R2 file). The relative reference (Ref. 58) was also included. Indeed, we make kindly note to the Reviewer, that an additive, as a cryoprotectant, is mandatory in the case of liposomes, due to the presence of an aqueous core which during the removal of solvent forms an empty inner space which can provoke the lipid collapse. This phenomenon can not occur in the case of micelles because they lack of an aqueous core, so that the addition of an additive would be nonsense. As confirmation to our statement, as an example we suggest to the Reviewer to see at: Rupa D. Dabholkar, Int J Pharm, 315 (2006), 148-157. In this study, as in our study, no additive was added and no filtration was performed but only a single dimensional family was detected during size analysis (Figure 1 of the study).

Reviewer 4 Report

Authors replies to the comments are satisfactory.

Author Response

We thank the Reviewer for his valuable suggestions and comments which helped us to significantly improve the quality of our manuscript